# Pharmacokinetics and Tissue Distribution of Coumarins from *Tagetes lucida* in an LPS-Induced Neuroinflammation Model

**DOI:** 10.3390/plants11212805

**Published:** 2022-10-22

**Authors:** Anislada Santibáñez, Maribel Herrera-Ruiz, Manasés González-Cortazar, Pilar Nicasio-Torres, Ashutosh Sharma, Enrique Jiménez-Ferrer

**Affiliations:** 1Centro de Investigación Biomédica del Sur, Instituto Mexicano del Seguro Social, Argentina No. 1 Col Centro, Xochitepec 62790, Mexico; 2School of Engineering and Sciences, Tecnologico de Monterrey, Av. Epigmenio González No. 500, San Pablo, Queretaro 76130, Mexico

**Keywords:** HPLC–UV, coumarins, *Tagetes lucida*, pharmacokinetics

## Abstract

*Tagetes lucida* has been widely used as a folk remedy in illnesses associated with the central nervous system and inflammatory ailments. Among the chemical compounds that stand out in the plant against these conditions are coumarins, such as 7-*O*-prenylscopoletin (PE), scoparone (SC), dimethylfraxetin (DF), herniarin (HR), and 7-*O*-prenylumbelliferone (PU), considered potential anti-neuroinflammatory compounds. Therefore, the relationship between the therapeutic effect and the dose can be evaluated through pharmacokinetic–pharmacodynamic (PK–PD) studies under a model of neuroinflammation induced by lipopolysaccharide (LPS). Nonetheless, accomplishing those studies requires an accurate and robust analytical method for the detection of these compounds in different biological matrices of interest. Due to the above, in the present study, a bioanalytical method was established by HPLC–DAD-UV for the simultaneous quantification of the coumarins present in the hexane extract of *T. lucida*, which was able to determine the temporal concentration profiles of each of the coumarins in the plasma, brain, kidney, and spleen samples of healthy and damaged mice. Coumarins showed an increase in plasma concentrations of up to three times in the neuroinflammation model, compared to healthy mice, so it was possible to quantify the therapeutic agents in the main target organ, the brain. The ability of compounds to cross the blood–brain barrier is an advantage in the treatment of diseases associated with neuroinflammation processes that can be studied in future PK–PD evaluations.

## 1. Introduction

*Tagetes lucida* Cav. of the Asteraceae family is a herb widely distributed in Mexico, well-known as “pericón”, “hierba anís”, “yauhtli”, or Mexican mint marigold [1]. Along with the use of *T. lucida*, as a ceremonial plant in some states of central Mexico, in traditional medicine, the leaves and flowers infusions are used as a folk remedy for gastrointestinal, respiratory, and inflammatory ailments, as well as an important appliance in illnesses associated with the central nervous system (CNS), such as “susto” and “nervios” (nervous sickness) [2].

Pharmacological activities have been evaluated with the essential oil and extracts of different polarities obtained from *T. lucida*. These effects have been related to the chemical content of the plant, where the phenolic compounds, such as coumarins, stand out [1,3,4,5,6]. Recent studies have proposed that 7-*O*-prenylscopoletin (PE), scoparone (SC), dimethylfraxetin (DF), herniarin (HR), and 7-*O*-prenylumbelliferone (PU), are the main coumarins characterized within the hexane, acetone, and aqueous extracts. Anti-inflammatory [5] anxiolytic [1], antidepressant [7], and vasorelaxant [6] activities have been reported for these compounds, which provides a reference for studies in more specific complaint states, such as neuroinflammation.

Neuroinflammation is a response to alterations in neuronal structure and function within the CNS caused by various peripheral inflammatory stimuli [8]. Neuroinflammatory long-state has been pointed out for its harmful effects on the development of neurodegenerative diseases [9]. Due to the complex scheme of the establishment and progression of the damage caused by neuroinflammation, combinations of therapeutic agents have been proposed in search of a treatment capable of stopping or reversing neuronal damage [10]. 

Due to its reproducibility, mice damaged with lipopolysaccharide (LPS) are one of the neuroinflammation models most widely used for the evaluation of anti-inflammatory phytochemicals over CNS [11]. This model could allow for a connection between pharmacokinetic–pharmacodynamic (PK–PD) studies for the anti-neuroinflammatory coumarins potential. Nonetheless, accomplishing those studies requires an accurate and robust analytical method for the detection of these types of compounds in different biological matrices.

Several studies of *T. lucida* have been focused mainly on pharmacological activities and phytochemical isolation and identification [12,13]. Additionally, high-performance liquid chromatography–diode array detection (HPLC–DAD) coupled with ultraviolet (UV) has been commonly used in the identification of coumarins from *T. lucida* extracts [1,3,5,14]. Nonetheless, a validated analytical method has not been designed to determine chemical composition diversity among the active extracts that can modify the pharmacological potency in live organisms by the interaction of these compounds. Anyway, the pharmacokinetic studies of these coumarins are null practically, and any study has evaluated the time-course concentration of these compounds with a potential neuroprotective effect. 

In the present study it was established and validated, based on FDA guidelines [15], a sensitive and cost-effective bioanalytical method by HPLC–DAD-UV to quantify five coumarins with potential neuroprotector effects, PE, SC, DF, HR, and PU, from hexanic extract of *T. lucida* in some of the main biological matrices used for in vivo preclinical studies: plasma, brain, kidney, and spleen. This quantification method allowed us to carry out a tissue distribution evaluation and a preliminary pharmacokinetic study that are fundamental requirements in monitoring therapeutic agents for the future establishment of PK–PD correlations in a neuroinflammatory state induced by LPS.

## 2. Results and Discussion

### 2.1. Optimization of Chromatographic Conditions

To our knowledge, an analytical method useful for the identification and quantification of the bioactive compounds obtained from the hexane extract of *T. lucida* has not been developed [1,3,4,5,7,14]. Therefore, the five coumarins, PE, SC, DF, HR, and PU, and the internal standard were injected under the liquid chromatographic system conditions to determine their retention time (RT), along with their UV spectra, shown in Figure 1. The system allowed for a suitable separation of the compounds from each for their further analysis in the validation process. Considering the UV spectrum of the analyzed compounds, maximum absorption wavelengths between 320 and 350 nm were observed. For this reason, a λ = 330 nm was used to identify and quantify all compounds in a single reading.

### 2.2. Sample Preparation

The optimal recovery of the interest compounds disseminated in these biological matrices is a critical step, since the presence of endogenous interferents must be avoided as much as possible. Liquid–liquid extraction (LLE) accompanied by previous protein precipitation (PP) is widely used, due to it guarantee regarding the elimination of interferences, as a result of the protein solubility decrease by the addition of organic solvents, such as methanol and acetonitrile, as well as its speed and feasibility, allowing for a better availability of the bioactive compounds present in the matrices [16].

In the present study, the PP was evaluated by adding methanol or acetonitrile to the plasma or tissue suspension. The PP with acetonitrile improved the sensitivity of the recovery method, although the recoveries were lower than the acceptance criteria established in the matrices evaluated. Then, a LLE step with dichloromethane was included, which guaranteed recovery above 85% of coumarins and IS in all cases of matrices. Finally, the supernatants of the centrifuged samples were evaporated and reconstituted in 100 µL of methanol to increase the coumarin concentration.

### 2.3. Method Validation

The validation of the bioanalytical method by HPLC–UV was completed, according to the recommendations established by the FDA guidance [15] for chromatographic assays.

#### 2.3.1. Specificity

The specificity was evaluated by comparing the chromatograms of blank matrices and blank matrices enriched with the analytes at the corresponding LLOQ concentrations, and plasma and tissue samples collected 15 min after oral administration of the bioactive fraction of *T. lucida* in the pharmacokinetic study (Figure 2). The analyzed samples did not show any interference in the matrices for the detection of PE (9.8 min), SC (11.4 min), DF (12.8 min), HR (13.7 min), PU (28.0 min), and IS (8.8 min).

#### 2.3.2. Linearity and Selectivity

The calibration curves were constructed from the ratio of the peak areas of the analytes and the IS. The curves were linear in all the biological matrices in a concentration range between 0.156–40 µg/mL for each coumarin and 1.5 µg/mL for IS. The equations that describe the linear regression models, the coefficients of determination (r^2^), and the specific LLOQs for each compound in the matrices are shown in Table 1. In the case of LLOQs, a single minimum concentration was established for all coumarins that correspond to 156 ng/mL, which meets the precision and accuracy acceptance criteria established above.

#### 2.3.3. Precision and Accuracy

The precision and accuracy estimated at the intra- and inter-day levels were evaluated in the five biological matrices by analyte mixed with the concentrations established for the low, medium, and high levels of the QC samples. The precision values for these levels were less than 11.05%, while the accuracy represented by %RE ranged from −14.01 to 8.66, varying between each matrix and analyte evaluated. All precision and accuracy values meet the criteria established by the FDA [15] and are summarized in Table 2.

#### 2.3.4. Recovery and Matrix Effect

Plasma and tissue matrices, spiked after the extraction process with QC levels, were compared to the previously spiked samples to determine the extraction recovery of both coumarin and IS. The lowest extraction value was 87.15% for PE in plasma (Table 3). Likewise, the matrix effect was evaluated with values between 85.09% and 107.77%, with RSD < 12.6%, for the different coumarins, while for the IS varied up to 7.59%, as shown in Table 3.

#### 2.3.5. Stability

The stability of the compounds obtained from each enriched matrices at the QC concentrations was evaluated under different conditions that covered the processing, handling, and storage phases that could be developed throughout the study. In all cases, the RE values showed that there were no significant losses of the analytes under short-term (autosampler) and long-term storage conditions, as well as in the freeze-thaw cycles of the samples, and data are summarized in Appendix A.

### 2.4. Pharmacokinetic and Tissue Distribution Study

The optimized HPLC-validated method for the quantification of coumarins present in the hexane extract of *T. lucida* was used to evaluate the concentration profiles over time in the plasma, brain, kidney, and spleen biological matrices after an oral administration of 10 mg/kg of the extract. 

Within PK and tissue distribution studies in animal models, whether healthy or diseased, it is important to look at the functional and structural capacities of each organ or system studied to obtain the most relevant information for the subsequent pharmacological studies or their later translation to humans. Therefore, in this study, each of these biological matrices offered a tool for the analysis of the dynamic behavior of compounds within the system [17].

Plasma presented a general scheme of the administration, distribution, metabolism, and excretion (ADME) processes and the bioavailability of the coumarins found in the hexane extract of *T. lucida* after its administration in both healthy mice and those subjected to LPS-induced neuroinflammation. Once the absorption process began, it was possible to determine if these compounds reached the target organ that corresponds to the brain and for how long they remained. In parallel, the elimination stage was compared with the amounts present in the kidneys to establish an association between excretion and systemic circulation. Finally, the presence of potential therapeutic agents in the spleen, one of the organs involved in the initial response to inflammatory processes, may clarify the results for subsequent pharmacodynamic studies, such as the evaluation of the presence of anti- and pro-inflammatory markers and its temporal relationship with coumarin concentrations, giving way to essential PK–PD studies in the development of new pharmacological therapies. The concentration mean time curves in the plasma of PE, SC, DF, HR, and PU from healthy and LPS-induced neuroinflammation mice were plotted in Figure 3. 

The behavior of the concentration variations over time between healthy and damaged subjects was similar in each of the coumarins. However, the graphic representation suggests that, in those mice that were previously administered with LPS, the amount of drug available in the plasma was three-fold higher than the value obtained in healthy subjects, as is the case of DF (Figure 3c) and HR (Figure 3d). Other studies have reported the impact of inflammation on pharmacokinetic variations, especially for plasma drug exposure by increasing the concentration/dose ratio [18]. 

Among the pharmacokinetic characteristics in the LPS-treated mice, the SC (Figure 3b) and DF (Figure 3c) highlight their fast distribution and subsequent elimination, consistent with that reported in previous studies [19,20]. The remaining coumarins show processes where the concentrations increase repeatedly or continuously, as is the case of PU (Figure 3e), which reaches a concentration peak up to 4 h after the administration of the *T. lucida* extract. This response should be considered for the design of the pharmaceutical forms used in further clinical studies, since its varied absorption may modify the effectiveness of some of the components. 

From these temporal profiles, the preliminary pharmacokinetic parameters shown in Table 4 were calculated for both conditions.

Mean C_max_ response in each coumarin was greater in mice damaged with LPS, and it was obtained with a shorter T_max_, so it was expected that the general exposure of each compound could be diversified among the animals. Except for SC, the AUC values showed a greater maintenance presence of therapeutic agents in plasma over time when the inflammatory agent was present, so theoretically, a system affected by an acute inflammation environment can increase the bioavailability of the therapeutic compounds in the bloodstream; then, it is important to consider adequate doses in pharmaceutical design to avoid intoxications or increased adverse effects, due to excess exposures [21].

ADME processes are known to be modified by the presence of inflammatory factors, such as pretreatment with agents, such as LPS [22]. Normally during the process of inflammation, absorption in the gastrointestinal tract can be modified by the presence of diarrhea as a result of the damage, thus modifying the distribution of therapeutic agents by changes in acute-phase plasma protein binding in response to inflammation. For instance, with the decrease in protein syntheses, such as albumin, and the increase in transferrin [23]. This may be related to the increase in the C_max_ of each of the coumarins in the damaged animals of up to three times more, as is the case with DF.

Additionally, it has been reported that the alterations in the hepatic and intestinal metabolism of drugs depend on factors such as blood flow rate, the free fraction of the drug in plasma, and the clearance rate defined by each compound. In pharmacokinetic studies of drugs metabolized mainly by the liver, the clearance rate is significantly slowed when individuals are subjected to LPS-mediated inflammatory processes [23]. The foregoing coincides with what is expressed in the observed values of oral clearance (Cl/F) in Table 4, where practically all the compounds studied presented a faster clearance in healthy systems (up to 31.4% increase), apart from SC, which maintained clearance rates close to each other.

Another of the main organs associated with drug clearance are the kidneys. When these are exposed to inflammatory states, renal excretion mediated by the glomerular filtration rate, as well as the plasma flow rate, shows a change in the pharmacokinetic profiles [18]. In our case, the net exposure of coumarins, detailed by AUC, increased and, in the case of SC, they remained comparable because of a slower excretion. The comparative results also showed an increase in serum concentration [23], compatible with the increase in plasma concentrations of the compounds observed in the present study.

Due to the profiles and pharmacokinetic parameters obtained, it was important to determine the distribution capacity of coumarins in the tissues. Figure 4 shows the distribution of the compounds in the brain, the main target organ, under conditions of systemic inflammation, compared to mice without damage. In turn, Appendix A set out the temporal distribution of coumarins in the kidneys and spleen, in addition to showing the transfer processes between the organs and the blood circulation system, the data obtained will be useful in subsequent PK–PD studies that evaluate the relevance of the pharmacological effects, concerning the processes of metabolism and clearance of drugs, for their therapeutic monitoring. 

In all cases, the brain was the organ with the lowest content of coumarins, below 5 ng/mL of the compounds in tissue, and this could be due to low penetration caused by the presence of the blood–brain barrier. Conversely, in comparison with a study where the distribution of SC in different organs was analyzed, it was reported that it was not possible to quantify this coumarin inside the brain [19], where the dose administered to the rats used for this study or the extraction process from the biosamples could be a limitation. 

The review carried out by Batista et al. (2019) detailed how the brain is affected in the LPS-induced systemic inflammation model, mentioning that, in addition to the presence of cytokines, such as interleukins IL-1β, IL-6, and TNF-α in plasma, there is evidence that LPS damages the blood–brain barrier, which allows for the infiltration of small therapeutic molecules, thus improving the pharmacological treatment [22]. 

The cooperation of the coumarins bioavailability in plasma (Figure 3) with the kinetic profiles in the brain (Figure 4) shows that the highest concentration in the tissue is reached 15 min after the initial absorption process of PE, SC, and PU coumarins. Moreover, HR and DF maintain similar behaviors in the plasma and brain, although in much lower concentrations for the latter. Factors such as the size of the molecules, as well as the highest in plasma, compared to the other compounds, influence this behavior.

In the distribution brain profile, an increase of the compounds up to three times more in the damaged mice is observed in the first 90 min, after this time, some coumarins, such as SC (Figure 4b) and PU (Figure 4e), maintain levels of higher tissue concentrations in control mice. It is possible that the neuroinflammation process by i.p. LPS administration caused damage to the blood–brain barrier, allowing for the permeability of these compounds, as previously described, which is an advantage in the treatment of diseases associated with neuroinflammation processes [11].

## 3. Materials and Methods

### 3.1. Reagents and Materials

Coumarins 7-*O*-prenylscopoletin (PE), scoparone (SC), dimethylfraxetin (DF), herniarin (HR), and 7-*O*-prenylumbelliferone (PU) were isolated and purified in the laboratory from a hexanic extract of *Tagetes lucida* (Figure 5). Each compound was identified by ^1^H- and ^13^C-NMR analyses, which were compared with those described in previous studies [5]. The internal standard rutin (IS, purity ≥ 98%), trifluoroacetic acid (TFA), and lipopolysaccharide (LPS) were purchased from Sigma-Aldrich (St. Louis, MO, USA); acetonitrile, methanol, and water solvents HPLC-grade from Tecsiquim (Mexico, Mexico), and reagent-grade hexane was purchased from Merck (Darmstadt, Germany).

### 3.2. Hexanic Extract Preparation

From a collection of *T. lucida* carried out in Xochitepec, Morelos, Mexico, the aerial parts of previously identified specimen material (Voucher No. 2081) [3] were dried on wire mesh beds at room temperature. The dry plant material was pulverized in a mill, until a particle size of 4–6 mm was obtained. Subsequently, 200 g of plant material was macerated three times in hexane for 24 h, each time the solvent was removed under reduced pressure. The standardization of the extract, in the content of coumarins, was determined by HPLC using the external standard method, calculating the following concentrations in mg/g of extract: 20.81 (PE), 14.64 (SC), 13.18 (DF), 19.79 (HR), and 20.35 (PU).

### 3.3. Animals

Male ICR mice (30 ± 5 g) were provided by the animal facility of XXI Century Medical Center, IMSS (CDMX, Mexico). The animals were kept at room temperature (22 ± 4 °C) with 12-h light-dark cycles (07:00 to 19:00 h). Access to food and water was allowed *ad libitum*, until 12 h before starting the experiment.

The studies were carried out following the Official Mexican Standard NOM-062-ZOO-1999: Technical Specifications for the Production, Care, and Use of Laboratory Animals [24]. This project was approved by the Local Committee for Research in Health and Ethics of the Mexican Institute of Social Security (IMSS) on 16 August 2021, with the registration number R 2021-1702-009.

### 3.4. Samples Collection

Blood samples were obtained from the retro-orbital sinus of mice and collected in heparinized tubes. Plasma was separated by centrifugation at 3500 rpm for 5 min and stored in new Eppendorf tubes at −70 °C, until further processing. 

After blood sampling, the mice were sacrificed in a chamber with chloroform by an overdose of anesthesia. The brain, kidney, and spleen were quickly removed and rinsed with saline solution, fresh weight was recorded, and immediately placed on ice. Subsequently, the organs were freeze-dried, grounded, and weighted. The lyophilized organs were individually suspended in methanol in a 1:1 volume: dry weight ratio for 24 h, then sonicated for 5 min and centrifuged at 14,000 rpm for 7 min. The supernatants were recovered in clean tubes and stored at −70 °C until their use. 

### 3.5. Preparation of Working Solutions, Calibration Curves, and Quality Control (QC) Samples

Individual stock solutions of PE, SC, DF, HR, PU, and rutin (IS) were prepared separately in methanol (1 mg/mL) and stored at −4 °C until use. From the stocks, a working solution in methanol was prepared at a concentration of 200 µg/mL of each coumarin. Blank plasma or organs supernatant samples (80 µL) were spiked with 20 µL of serial dilutions of working solution to obtain calibration curves ranging between 0.156, 0.312, 0.625, 1.25, 2.5, 5, 10, 20, and 40 µg/mL of each coumarin. Quality control (QC) samples were prepared at three levels of 0.3 (low), 3 (medium), and 30 (high) µg/mL for each coumarin.

### 3.6. Plasma and Tissue Samples Processing

One hundred microliters of plasma and organs samples, spiked with calibration and QC concentration or samples from the pharmacokinetic study, were mixed with 300 µL of acetonitrile containing the IS at 10 µg/mL for protein precipitation, and vortexed for 3 min. Then, 200 µL of dichloromethane were added, for liquid–liquid extraction of the bioactive compounds, and vortexed for 5 min. The mixture was centrifuged at 14,000 rpm for 10 min, and the extracted organic layers were placed in new tubes, until completely dry at room temperature. For quantitative analysis, samples were resuspended in 100 µL of methanol, transferred to sampling vials, and injected into the chromatographic system, described below. 

### 3.7. HPLC–DAD-UV Handling Conditions

Biological and standard samples analysis was carried out by a high-performance liquid chromatography Waters 2695 series. The Waters 2995 series HPLC separation module consisted of a quaternary pump, degasser, autosampler, and thermostatted column. Additionally, it was connected to a photodiode array UV–VIS detector, Waters 2996 series. A Supelco Discovery^®^ C18 column (250 × 4.6 mm, 5 µm, Merck) was used for chromatographic separation and method validation. The injection volume of all described samples was 10 µL, using a mobile phase flow set at 0.9 mL/min, consisting of a 0.5% trifluoroacetic acid aqueous solution (A) and acetonitrile (B). The final run time of the samples was established at 30 min, with a gradient solution as follows: 0–2 min, 100–95% (A); 2–4 min, 95–70% (A); 4–21 min, 70–50% (A); 21–24 min, 50–20% (A); 24–27 min, 20–0% (A); and 27–30 min, 0–100% (A). The readings were carried out at a wavelength (λ) of 330 nm and processed with the Empower Pro 3.0 software (Waters, MA, USA).

### 3.8. Validation of HPLC–DAD-UV Quantification Method

The validation of the HPLC quantification method of the coumarins present in the *T. lucida* hexanic extract was carried out under the FDA Bioanalytical Method Validation: Guidance for Industry [15].

#### 3.8.1. Specificity, Linearity, and Sensitivity

Specificity was ensured by evaluating that there were no endogenous interferents in the working matrix corresponding to blank biological matrices (plasma and tissues) from six mice. Linearity was determined by linear regression of the calibration curves, based on the ratio of the area under the curve (AUC) of the analyte peak against the same IS signal. Sensitivity was defined as the lowest concentration that can be determined, according to the signal/noise (S/N) ≥ 5 of the analyte peaks, obtained by the lower limit of quantification (LLOQ). The evaluation of the LLOQ must meet the acceptance criteria of accuracy and precision, which are evaluated with relative standard deviation (RSD) ≤ 20%.

#### 3.8.2. Extraction Recovery and Matrix Effects

Recovery was analyzed in the concentrations of the QC samples by comparing the peak areas of samples extracted under the normal procedure with the post-extraction spiked samples at the same concentration levels (*n* = 5). On the other hand, the effect of the matrix on the analytical evaluation of the coumarins was obtained by comparing five samples extracted from plasma and each organ against five samples prepared directly in the mobile phase. The matched concentrations corresponded to the three QC points, including the LLOQ established before. The effect of the matrix was calculated by means of the coefficients of the AUC of samples extracted from the matrices and those prepared in the system, with respect to the response signal of the IS for each coumarin. The nominal concentration obtained in the processed matrices must range between ±15%, evaluated by relative error (RE) and ≤15% for RSD.

#### 3.8.3. Precision and Accuracy

Intra- and inter-day precision and accuracy were determined by evaluating six replicates of QC samples, prepared independently from three different sets. The acceptance criteria corresponded to RSD ≤ 15% and RE ± 15%, according to the nominal concentration.

#### 3.8.4. Stability

The stability of the analytes in plasma, brain, kidneys, and spleen was evaluated by the analyzed concentrations of QC subjected to three different conditions. Short-term stability was determined by analysis of samples processed in the autosampler vials at 24 h at room temperature. Long-term stability was tested on samples stored at −70 °C for 30 days. Freeze-thaw stability was determined after three cycles (−4 °C to 25 °C) on three consecutive days.

### 3.9. LPS-Induced Neuroinflammation and Pharmacokinetic Study

A pharmacokinetic study was carried out in sixty mice, divided in six subgroups, according to sample collection times. An acute inflammation process by the intraperitoneal (i.p.) administration of LPS at 2 mg/kg was induced in half of the mice per group, while the other half received an i.p. injection of saline solution. Ten min after the damage was induced, the mice were administered with an oral dose of 10 mg/kg of the bioactive fraction dissolved in a 1% Tween-20 aqueous solution. Blood and tissue samples were obtained and processed at 0, 0.25, 0.75, 1.5, 2, 4, and 6 h post-dosing, as indicated in the “Samples collection” and “Plasma and tissue samples processing” sections.

### 3.10. Pharmacokinetic and Tissue Distribution Analysis

Pharmacokinetic parameters of PE, SC, DF, HR, and PU in plasma were calculated by PKSolver software [25]. The maximum plasma concentration (C_max_), time to reach the maximal concentration (T_max_), half-life time (t_1/2_), area under the concentration–time curve to 6 h (AUC_0-6_) and infinity (AUC_0-∞_), mean residence time (MRT), and the observed oral clearance (Cl/F) were obtained using a non-compartmental model, expressed as mean ± SEM.

To evaluate the distribution in the tissues, bioactive coumarins were quantified in the lyophilized organs, and concentrations were adjusted to the volumes of liquid obtained by the difference between the dry and fresh weights of each tissue to simulate a distribution approximation of the compounds by tissue system.

## 4. Conclusions

A sensitive, suitable, and validated HPLC–DAD-UV method for the simultaneous quantification of five coumarins, i.e., PE, SC, HR, DF, and PU, in the plasma, brain, kidneys, and spleen was developed and successfully applied in preclinical pharmacokinetic and tissue distribution studies, following an oral administration of hexanic extract of *Tagetes lucida* in healthy and damaged mice by an LPS-induced neuroinflammation model. The bioavailability observed in brain tissue and plasma determined that the compounds could reach the target site to exert their potential therapeutic functions in systems damaged by the agents that cause neuroinflammation. The present study has potential applicability for further pharmacokinetic–pharmacodynamic evaluations to determine the correlation between different dose administrations and their therapeutic effects on central nervous system ailments.

## Figures and Tables

**Figure 1 plants-11-02805-f001:**
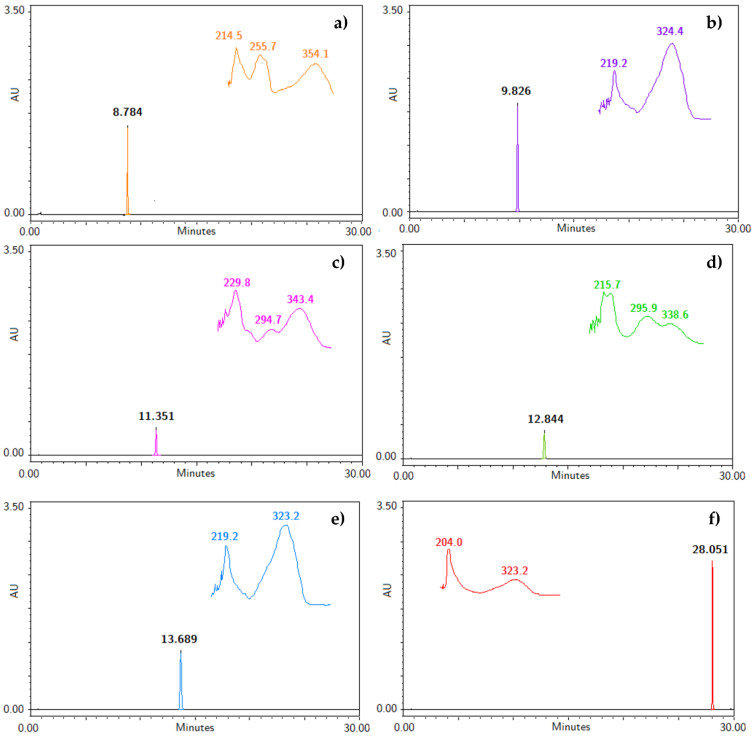
Chromatograms with retention time (RT) and specific UV spectra of (**a**) IS, (**b**) PE, (**c**) SC, (**d**) DF, (**e**) HR, and (**f**) PU at a concentration of 200 µg/mL.

**Figure 2 plants-11-02805-f002:**
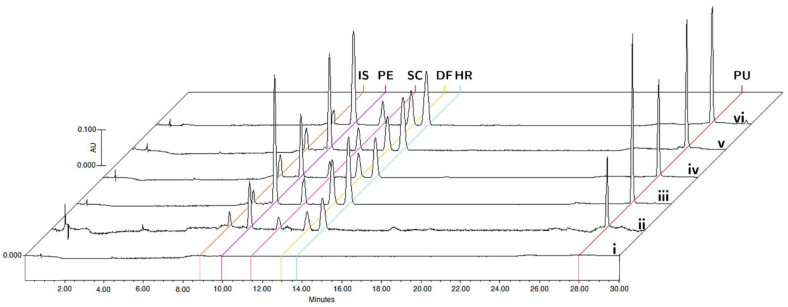
Representative chromatograms of (i) blank matrix, (ii) blank matrix spiked with LLOQ concentration, and processed samples collected at 15 min after oral administration of standardized hexane extract spiked with IS in plasma (iii), brain (iv), kidneys (v), and spleen (vi).

**Figure 3 plants-11-02805-f003:**
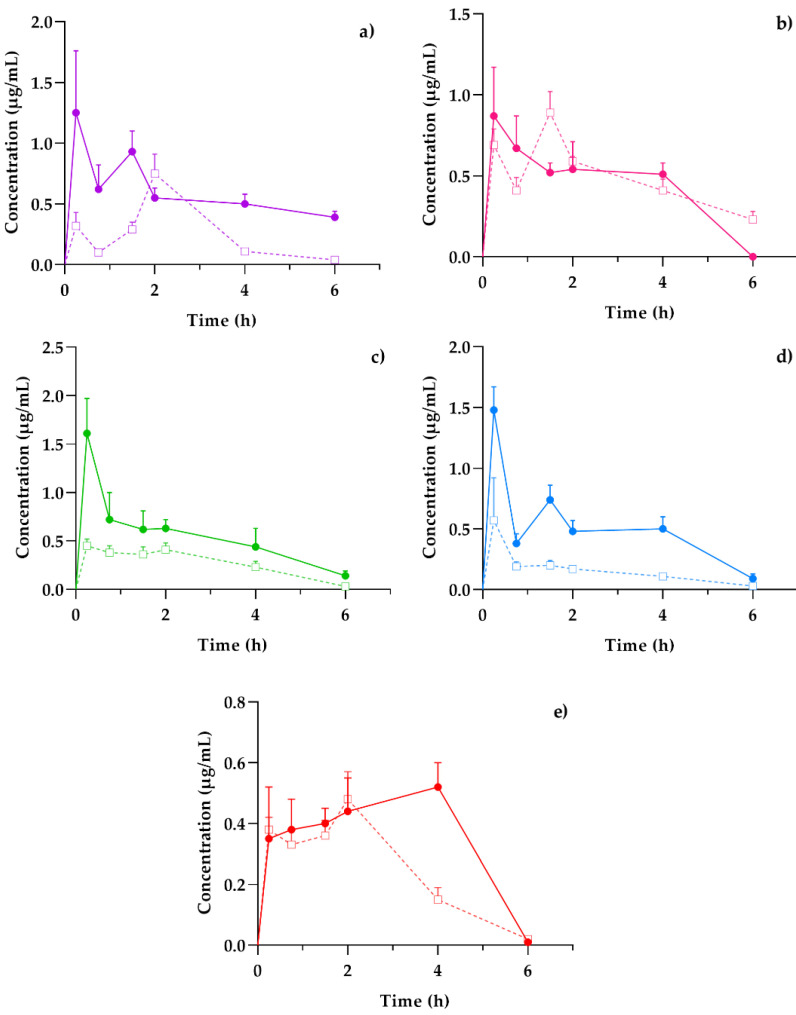
Concentration–time profiles of coumarins: (**a**) PE, (**b**) SC, (**c**) DF, (**d**) HR, and (**e**) PU after a 10 mg/kg oral dose administration of hexanic extract of *Tagetes lucida* in healthy (-□-) and LPS-damaged (–•–) ICR mice. The results are presented as mean ± SEM (*n* = 5).

**Figure 4 plants-11-02805-f004:**
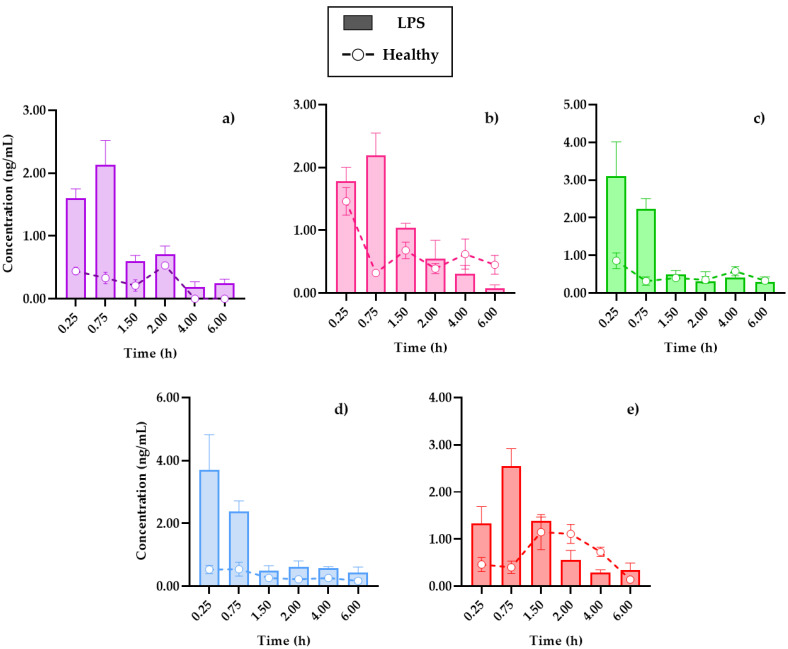
Distribution in the brain of (**a**) 7-*O*-prenylscopoletin (PE), (**b**)scoparone (SC), (**c**) dimethylfraxetin (DF), (**d**) herniarin (HR), (**e**) 7-*O*-prenylumbelliferone (PU) after an oral dose administration of hexanic extract of *Tagetes lucida* in healthy and LPS-administrated mice. Values are presented as mean ± SEM (*n* = 5).

**Figure 5 plants-11-02805-f005:**
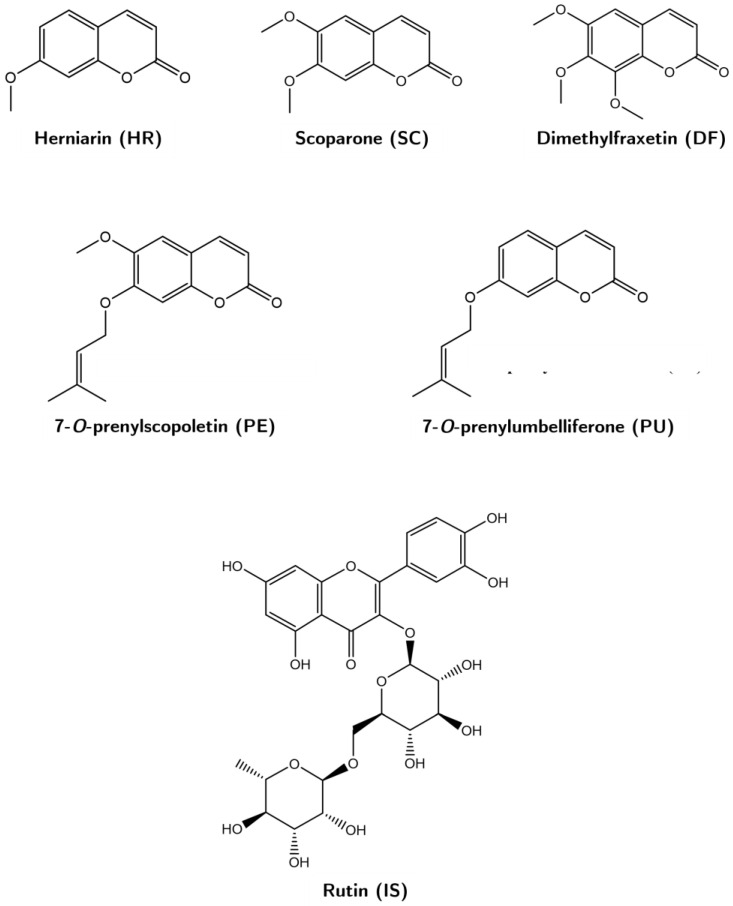
Chemical structures of the coumarins 7-*O*-prenylscopoletin (PE), scoparone (SC), dimethylfraxetin (DF), herniarin (HR), and 7-O-prenylumbelliferone (PU), and of the rutin (IS).

**Table 1 plants-11-02805-t001:** Linearity equations of calibration curves and LLOQ determined for coumarins in plasma and tissue matrices.

Matrix	Analytes	Linearity Equation	r^2^	LLOQ (µg/mL)
Plasma	PE	y = 0.2278x + 0.0246	0.9996	0.18
	SC	y = 0.0743x − 0.0126	0.9987	0.05
	DF	y = 0.1260x − 0.0322	0.9965	0.15
	HR	y = 0.2082x + 0.0294	0.9997	0.17
	PU	y = 0.2705x + 0.0916	0.9976	0.10
Brain	PE	y = 0.1607x + 0.0046	0.9995	0.13
	SC	y = 0.0510x − 0.0094	0.9991	0.10
	DF	y = 0.0841x − 0.0037	0.9997	0.08
	HR	y = 0.1455x − 0.0043	0.9969	0.10
	PU	y = 0.1886x + 0.0275	0.9997	0.02
Kidney	PE	y = 0.2394x + 0.024	0.9998	0.07
	SC	y = 0.0746x − 0.0054	0.9995	0.18
	DF	y = 0.1273x − 0.0119	0.9994	0.03
	HR	y = 0.2104x − 0.0376	0.9986	0.16
	PU	y = 0.2794x − 0.0397	0.9993	0.02
Spleen	PE	y = 0.3918x + 0.0278	0.9996	0.12
	SC	y = 0.1238x − 0.0284	0.9986	0.02
	DF	y = 0.2090x − 0.0218	0.9991	0.05
	HR	y = 0.3603x + 0.0271	0.9989	0.11
	PU	y = 0.4728x − 0.1095	0.9989	0.02

PE = 7-*O*-prenylscopoletin, SC = scoparone, DF = dimethylfraxetin, HR = herniarin, and PU = 7-*O*-prenylumbelliferone.

**Table 2 plants-11-02805-t002:** Data of accuracy and precision determined for coumarins in plasma and tissue matrices.

Nominal Concentrations (µg/mL)		30.0				3.0				0.3			
Matrix	Analytes	Repeatability		Reproducibility		Repeatability		Reproducibility		Repeatability		Reproducibility	
		RE (%)	RSD (%)	RE (%)	RSD (%)	RE (%)	RSD (%)	RE (%)	RSD (%)	RE (%)	RSD (%)	RE (%)	RSD (%)
**Plasma**	**PE**	1.30	1.73	1.05	1.49	1.44	1.79	1.71	1.34	8.66	1.49	4.59	4.83
	**SC**	−5.28	2.41	−4.94	1.73	−9.61	3.74	−8.25	3.04	−6.67	3.55	−12.71	6.69
	**DF**	−11.95	1.45	−7.80	5.19	−7.85	1.08	−5.39	2.93	−14.01	6.30	−10.64	5.7
	**HR**	−6.69	1.51	−2.56	4.74	1.31	1.03	1.84	1.02	−2.18	2.83	0.06	3.07
	**PU**	1.96	1.66	1.94	1.27	−0.17	0.90	0.06	0.75	−5.66	2.11	−4.36	2.02
**Brain**	**PE**	0.99	4.14	−0.12	3.10	5.34	1.58	2.96	2.76	2.09	1.28	2.37	1.94
	**SC**	−0.64	1.82	−1.81	2.19	3.97	0.61	1.96	2.39	−6.41	5.35	−7.98	5.41
	**DF**	1.97	3.71	1.54	2.46	7.81	2.78	3.17	7.07	−4.03	2.36	−2.68	3.77
	**HR**	1.77	3.74	1.17	3.47	−1.79	3.70	−5.53	5.10	2.78	1.35	3.69	1.62
	**PU**	3.47	1.98	1.49	2.75	4.32	0.92	−0.98	6.28	−5.11	6.81	−7.60	5.42
**Kidney**	**PE**	−3.59	2.30	−2.08	2.35	0.94	0.23	0.51	1.06	−2.02	1.21	−1.46	1.06
	**SC**	−0.65	2.03	−0.90	1.45	2.39	1.04	1.35	1.51	0.07	1.27	0.20	0.86
	**DF**	−0.27	1.23	−0.20	0.90	−2.18	6.29	−1.81	4.08	−0.07	3.08	−1.08	4.33
	**HR**	3.58	3.12	2.10	2.99	−4.14	2.43	−0.26	5.07	−6.52	3.33	−7.12	2.66
	**PU**	−0.21	2.02	−0.15	1.90	−4.41	6.51	−1.66	5.24	−5.93	6.14	−4.70	5.71
**Spleen**	**PE**	1.64	1.30	1.90	3.04	−10.29	6.93	−4.80	8.00	−4.52	2.50	−8.01	9.87
	**SC**	−0.43	7.32	−2.54	11.05	−12.42	3.32	−11.73	6.25	−6.69	8.82	−7.46	8.03
	**DF**	−9.08	7.89	−10.04	5.33	−4.93	10.71	−5.10	8.77	−3.77	9.44	−4.16	7.76
	**HR**	2.34	2.65	1.47	2.95	−3.52	9.67	−2.20	6.82	−2.38	5.45	−5.44	6.58
	**PU**	−7.17	4.30	−6.18	3.39	−10.72	2.70	−8.36	7.00	−8.99	2.94	−4.00	6.55

PE = 7-*O*-prenylscopoletin, SC = scoparone, DF = dimethylfraxetin, HR = herniarin, and PU = 7-*O*-prenylumbelliferone.

**Table 3 plants-11-02805-t003:** Data of extraction recovery and matrix effects determined for coumarins in plasma and tissue matrices.

Spiked Concentrations (µg/mL)		30.0				3.0				0.3			
Matrix	Analytes	Extraction Recovery		Matrix Effect		Extraction Recovery		Matrix Effect		Extraction Recovery		Matrix Effect	
		RE (%)	RSD (%)	RE (%)	RSD (%)	RE (%)	RSD (%)	RE (%)	RSD (%)	RE (%)	RSD (%)	RE (%)	RSD (%)
**Plasma**	**PE**	99.04	7.11	94.60	7.78	87.15	8.25	89.72	9.55	88.00	10.41	86.27	10.43
	**SC**	98.30	2.38	98.55	7.53	91.67	8.16	92.52	9.53	94.13	5.78	89.59	9.33
	**DF**	102.04	3.91	101.23	5.45	100.88	10.35	94.83	11.26	97.58	7.33	102.99	4.70
	**HR**	99.67	6.21	94.79	6.08	87.83	7.20	92.93	8.53	93.82	3.83	99.70	6.13
	**PU**	102.17	8.05	94.66	7.11	94.77	1.27	85.09	3.10	101.67	9.42	108.91	10.23
	**IS**	99.42	4.66	99.61	7.59	94.22	2.43	92.24	2.38	99.71	3.77	100.59	3.66
**Brain**	**PE**	91.64	4.80	106.08	6.48	88.01	0.54	102.28	2.64	92.19	1.21	98.72	3.11
	**SC**	99.44	1.23	104.99	7.68	94.65	1.06	100.94	4.21	89.70	6.03	94.89	4.67
	**DF**	90.13	1.60	107.26	5.22	96.44	5.16	105.50	7.29	93.08	3.45	94.55	5.75
	**HR**	91.81	3.22	105.11	4.90	94.27	3.10	104.20	5.42	112.29	1.54	100.92	3.04
	**PU**	93.49	5.98	107.77	7.35	99.56	3.95	101.16	9.57	95.23	5.72	96.32	9.37
	**IS**	95.11	2.75	95.32	6.31	98.97	2.28	99.01	2.81	99.84	2.65	94.71	3.90
**Kidney**	**PE**	97.95	2.29	96.77	2.67	99.28	0.93	96.08	6.18	91.30	1.16	100.34	4.79
	**SC**	99.99	2.80	98.86	1.70	105.72	0.97	95.55	5.96	94.28	9.05	100.39	12.60
	**DF**	99.40	2.30	97.20	2.81	104.84	4.78	98.51	7.05	87.24	2.72	96.97	9.83
	**HR**	103.06	6.55	100.89	4.06	91.38	2.42	95.68	6.09	90.39	1.25	99.26	5.03
	**PU**	100.87	8.68	98.52	5.28	103.27	5.16	97.70	10.57	93.11	3.89	97.25	6.83
	**IS**	101.26	1.25	101.28	1.63	103.72	3.20	103.89	5.66	99.94	2.75	109.67	4.86
**Spleen**	**PE**	98.33	11.85	99.87	6.05	100.55	1.93	100.36	2.92	90.36	1.02	100.04	4.43
	**SC**	100.16	3.04	101.09	5.38	109.71	1.28	99.54	1.78	88.30	7.77	103.43	9.26
	**DF**	100.77	6.28	101.70	5.41	104.20	2.71	98.98	5.98	92.01	6.31	105.13	7.23
	**HR**	100.99	10.13	101.12	6.77	92.58	2.45	98.66	3.54	88.74	1.56	101.12	3.64
	**PU**	99.31	14.67	98.83	3.34	105.05	4.03	102.63	6.70	91.89	4.61	99.32	4.65
	**IS**	98.87	3.19	98.96	4.87	99.98	0.94	100.00	1.73	99.02	2.74	97.02	2.78

PE = 7-*O*-prenylscopoletin, SC = scoparone, DF = dimethylfraxetin, HR = herniarin, and PU = 7-*O*-prenylumbelliferone.

**Table 4 plants-11-02805-t004:** Pharmacokinetic parameters of coumarins estimated by non-compartmental analysis in plasma after oral administration of hexanic extract of *Tagetes lucida* in healthy and LPS-damaged ICR mice.

Parameter	Unit	PE		SC		DF		HR		PU	
		Healthy	LPS	Healthy	LPS	Healthy	LPS	Healthy	LPS	Healthy	LPS
**C_max_**	µg/mL	0.93 ± 0.29	1.77 ± 0.31	0.97 ± 0.10	1.28 ± 0.11	0.58 ± 0.02	1.82 ± 0.27	0.62 ± 0.34	1.48 ± 0.19	0.56 ± 0.05	0.69 ± 0.07
**T_max_**	h	1.55 ± 0.34	0.85 ± 0.28	1.25 ± 0.25	0.70 ± 0.34	1.20 ± 0.31	0.60 ± 0.24	0.95 ± 0.35	0.25 ± 0.00	1.55 ± 0.23	1.05 ± 0.31
**t_1/2_**	h	1.86 ± 0.55	4.48 ± 1.31	2.88 ± 0.83	0.65 ± 0.06	0.98 ± 0.27	1.50 ± 0.37	1.66 ± 0.46	1.49 ± 0.35	0.92 ± 0.06	0.95 ± 0.10
**AUC_0-t_**	µg·h/mL	1.56 ± 0.35	3.52 ± 0.10	2.87 ± 0.09	2.77 ± 0.21	1.64 ± 0.17	3.25 ± 0.31	0.92 ± 0.12	2.95 ± 0.28	1.49 ± 0.11	2.22 ± 0.26
**AUC_0-∞_**	µg·h/mL	1.70 ± 0.31	6.37 ± 1.20	4.03 ± 0.63	2.78 ± 0.21	1.71 ± 0.19	3.61 ± 0.40	1.03 ± 0.14	3.21 ± 0.38	1.52 ± 0.12	2.24 ± 0.26
**MRT**	h	3.04 ± 0.45	6.96 ± 1.92	4.68 ± 1.16	2.26 ± 0.17	2.35 ± 0.33	2.72 ± 0.30	2.77 ± 0.35	2.77 ± 0.25	2.15 ± 0.11	2.68 ± 0.14
**Cl/F**	*	65.83 ± 10.18	17.38 ± 2.25	26.64 ± 3.00	36.52 ± 2.68	62.45 ± 9.38	28.81 ± 2.51	103.7 ± 13.0	32.50 ± 2.91	67.37 ± 5.33	46.95 ± 5.03

Values represent mean ± SEM (*n* = 5). PE = 7-*O*-prenylscopoletin, SC = scoparone, DF = dimethylfraxetin, HR = herniarin, and PU = 7-*O*-prenylumbelliferone. * (mg/kg)/(μg/mL)/h.

## Data Availability

The data presented in the study are available in the article and its Appendix A.

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
