# Peer review of "Pharmacokinetics and Tissue Distribution of Coumarins from Tagetes lucida in an LPS-Induced Neuroinflammation Model"

_plants, 2022, doi:10.3390/plants11212805_

Round 1
Reviewer 1 Report
The purpose of this manuscript is to evaluate the relationship between the therapeutic effect of coumarin and the dose through a pharmacokinetic-pharmacodynamic (PK-PD) study in a lipopolysaccharide-induced neuroinflammation model. The authors established a bioassay and observed coumarin bioavailability in brain tissue and plasma, confirming that coumarin not only reaches its target site, but also improves the effects of drugs that cause neuroinflammation. Interestingly, although LPS administration causes damage to the blood-brain barrier, it also makes these compounds permeable and improves the efficacy of drug therapy. I consider this study complete enough to be accepted into this journal.
Author Response
General Comment:
The purpose of this manuscript is to evaluate the relationship between the therapeutic effect of coumarin and the dose through a pharmacokinetic-pharmacodynamic (PK-PD) study in a lipopolysaccharide-induced neuroinflammation model. The authors established a bioassay and observed coumarin bioavailability in brain tissue and plasma, confirming that coumarin not only reaches its target site, but also improves the effects of drugs that cause neuroinflammation. Interestingly, although LPS administration causes damage to the blood-brain barrier, it also makes these compounds permeable and improves the efficacy of drug therapy. I consider this study complete enough to be accepted into this journal.
Response:
Dear reviewer, we really appreciate your comments. Thank you for your review.
Reviewer 2 Report
Manuscript plants-1974879 entitled “Pharmacokinetics and tissue distribution of coumarins from Tagetes lucida in an LPS-induced neuroinflammation model” developed and validated an HPLC-UV method for quantifying multiple types of coumarins from Tagets lucida in mouse. The manuscript is well-written. And there are some points need to be corrected or clarified before publication:
1. In section 3.4, it was indicated the organs were frozen after collection without wash. Therefore, the accuracy of the measured results is concerned as potential blood contamination of target compounds to organs without wash might happen during the collection stage.
2. According to FDA guideline, low QC level should be 3X of LLOQ, however, the concentration of LQC for target compounds in different biomatrices were all set at 0.3 μg/mL. Please explain it.
3. In section 2.3.2, the authors indicated “The equations that describe the linear regression models, coefficients of determination (r2), as well as the specific LLOQs for each compound in the matrices are shown in Table 1. In the case of LLOQs, a single minimum concentration was established for all coumarins that correspond to 156 ng/mL which meets the precision and accuracy acceptance criteria established above.” Please make a good explanation of setting a universal LLOQ conc. (i.e. 156 ng/mL) for all compounds in different matrices. According to FDA guideline, LLOQ is the first non-zero calibrator of the quantification curve.
4. In Section 2.1, the authors should indicate the reason of choosing 330 nm as the detection wavelength.
Author Response
Manuscript plants-1974879 entitled “Pharmacokinetics and tissue distribution of coumarins from Tagetes lucida in an LPS-induced neuroinflammation model” developed and validated an HPLC-UV method for quantifying multiple types of coumarins from Tagetes lucida in mouse. The manuscript is well-written. And there are some points need to be corrected or clarified before publication:
Comment 1:
In section 3.4, it was indicated the organs were frozen after collection without wash. Therefore, the accuracy of the measured results is concerned as potential blood contamination of target compounds to organs without wash might happen during the collection stage.
Response 1:
Dear reviewer, thank you for your advice. We omitted to write that the organs were rinsed with saline solution, therefore the comment was added to the document:
Section 3.4: “The brain, kidney, and spleen were quickly removed and rinsed with saline solution, recording the fresh weight, and immediately placed on ice.”
Comment 2:
According to FDA guideline, low QC level should be 3X of LLOQ, however, the concentration of LQC for target compounds in different biomatrices were all set at 0.3 μg/mL. Please explain it.
Response 2:
Thank you for your comment. Although the FDA guidelines define that the lowest quality control is 3x the LLOQ, this corresponds to a recommendation, but it is not mandatory, so we chose this value to broaden the quality control range that met the accuracy and precision criteria.
Comment 3:
In section 2.3.2, the authors indicated “The equations that describe the linear regression models, coefficients of determination (r2), as well as the specific LLOQs for each compound in the matrices are shown in Table 1. In the case of LLOQs, a single minimum concentration was established for all coumarins that correspond to 156 ng/mL which meets the precision and accuracy acceptance criteria established above.” Please make a good explanation of setting a universal LLOQ conc. (i.e. 156 ng/mL) for all compounds in different matrices. According to FDA guideline, LLOQ is the first non-zero calibrator of the quantification curve.
Response 3:
Values recorded in Table 1 were established in accordance with the sensitivity acceptance criteria where “the analyte response at the LLOQ should be ≥ five times the analyte response of the zero calibrator”. Nonetheless, methodologically the concentration 156 ng/mL was selected as a consensus value included in the LLOQ of the different matrices to avoid work with 25 different LLOQs and optimize the ranging level lecture.
Comment 4:
In Section 2.1, the authors should indicate the reason of choosing 330 nm as the detection wavelength.
Response 4:
We acknowledge your comment. Now we have stated the argument for wavelength choosing.
Section 2.1: “Considering the UV spectrum of the analyzed compounds, maximum absorption wavelengths between 320 and 350 nm were observed. For this reason, a λ= 330 nm was used to identify and quantify all compounds in a single reading.”